# Microbiologic Analysis of Removed Silicone Punctal Plugs in Dry Eye Patients

**DOI:** 10.3390/jcm11092326

**Published:** 2022-04-21

**Authors:** Il Jung, Jung Suk Yoon, Byung Yi Ko

**Affiliations:** 1Department of Ophthalmology, Konyang University College of Medicine, Daejeon 35365, Korea; jungil@kyuh.ac.kr; 2Department of Ophthalmology, Seoul National University Hospital, Seongnam 13620, Korea; jungseok91@gmail.com

**Keywords:** silicone punctal plug, culture, dry eye

## Abstract

This study analyzed the microbiologic results of removing silicone punctal plugs due to uncomfortable symptoms in dry-eye patients. Patients who were diagnosed with dry eye and received silicone punctal plugs—SuperEagle Punctum Plug™ (EagleVision, Denville, NJ, USA) or Parasol Punctum Plug™ (Beaver–Visitec international, Inc., Waltham, MA, USA)—into upper or lower puncta that were removed due to discomfort from January 2018 to June 2020 were enrolled and reviewed retrospectively. Out of the total 58 patients (64 eyes), 19 patients were male and 39 patients were female. Protrusion without granulation (21 patients, 32.8%) was the most common reason for plug removal, followed by protrusion with granuloma (19 patients, 29.7%). The positive rate of bacterial culture was 42.2% and *Klebsiella aerogenes* was the most common organism identified (18.5%). Vancomycin showed the highest susceptibility of 100% among all the antibiotics, third-generation cephalosporins were the most susceptible (88.5%) among cephalosporines, and levofloxacin was the most susceptible (81.0%) among quinolones. Among the patients who complained of discomfort after insertion of silicone punctal plugs, approximately 42% had a positive result in bacterial culture. Therefore, when removing punctal plugs in such patients, a microbiological examination may be needed for the appropriate selection of antibiotics.

## 1. Introduction

Punctal plug insertion is an important treatment method for dry eye associated with a severe aqueous deficiency or connective tissue disease. This method is known to occlude the punctum to preserve tears and therefore facilitate the enhancement of both tear film quality and quantity [1,2,3,4,5]; it has also been reported to improve dry-eye symptoms and reduce the need for eye drops by improving the ocular surface [6]. The first punctal plug was developed and used by Freeman in 1975 [7], and the silicone punctal plug has since been widely used in clinical practice. However, punctal plug insertion has also been reported to cause complications such as epiphora, corneal or conjunctival abrasion, plug extrusion, granuloma formation, and punctal stenosis [8,9,10,11,12], and this method is not equally effective for all dry-eye patients. Thus, the selection of punctal plugs of the right type and size and their usage in the appropriate patient group is critical.

Punctal plugs can also promote the formation of biofilms, which consist of bacteria and extracellular matrices produced by bacteria [13] and are known to cause eye infections that are difficult to treat due to their high rates of recurrence and resistance to human defense mechanisms. Biofilms are widely known to cause infections related to biological materials, especially in urethral catheters, venous catheters, and prosthetic heart valves [14,15]. In the field of ophthalmology, biofilms have been observed on the contact lens surface of patients with infectious keratitis in association with contact lens usage and on the surface of the scleral buckle in patients with retinal detachment after scleral buckling. Biofilms have also been reported in patients with endophthalmitis caused by bacteria after cataract surgery [16,17,18,19].

Bacterial biofilms may also form in punctal plugs. In the study by Sugita et al. [20], the presence of bacteria and bacterial biofilms in the punctal plug was confirmed using culture tests, scanning electron microscopy, and transmission electron microscopy. In addition, Ratheesh at al. quantitatively measured the thickness of biofilms using optical coherence tomography [21,22]. Previous studies have reported that biofilms in the punctal plug may cause bacterial infection in the eyeball or adnexa [20], and bacterial biofilms in the punctal plug have been shown to cause acute conjunctivitis [23], with biofilms likely to be the cause of discomfort related to the punctal plug. Thus, we conducted a microbiological test on the plug upon its removal because of the symptoms of discomfort after the insertion of silicone punctal plug in patients with dry eye, and analyzed the test results.

## 2. Materials and Methods

We retrospectively analyzed the medical records of 58 patients (64 eyes) who underwent microbiological tests after removal of a punctal plug due to discomfort related to plug extrusion, granuloma formation, inflammation, epiphora, or irritation. The patients had been diagnosed with dry eye based on the guidelines of the Korean Corneal Disease Study Group [24] during their visit to the Department of Ophthalmology at Konyang University Hospital between January 2018 and June 2020 and subsequently underwent insertion of either the SuperEagle Punctum Plug™ (EagleVision, Denville, NJ, USA) or Parasol Punctum Plug™ (Beaver–Visitec International, Inc., Waltham, MA, USA). This study was conducted in compliance with the Declaration of Helsinki and was approved by the Institutional Review Board of Konyang University Hospital (KYUH 2020-10-007).

The demographic variables were patient age, sex, Sjögren’s syndrome, diabetes, hypertension, and anterior segment eye disease. The clinical variables, which were retrospectively investigated, included the number of patients showing a positive result in the bacterial culture test, the cause of removal of the punctal plug, the retention period after punctal plug insertion, the lower or upper punctum of punctal plug insertion, the causal bacteria, the results of Gram staining, and the antibiotic susceptibility.

For microbiological testing, smear and culture tests were performed using the punctal plug removed from the patient’s eye. The first step in the smear test was topical anesthesia of the ocular surface with 0.5% proparacaine (Alcaine^®^, Alcon laboratory, Fort Worth, TX, USA), after which the punctal plug was gently pulled away from the lacrimal punctum by using sterile forceps with simultaneous observation using slit-lamp microscopy. During this process, care was taken to avoid contact of the plug with the eyelashes or other tissues. After removal, a smear specimen of the punctal plug was placed on a glass slide, and the results of Gram staining were obtained. In the culture test, the samples were inoculated onto blood agar and chocolate agar. The cultured bacteria were identified using an automatic microbiological analyzer (Microscan Walkaway 96 Plus system; Siemens Healthcare Diagnostics, West Sacramento, CA, USA).

For the antibiotic-susceptibility test, the Kirby–Bauer disc-diffusion method [25] was used in addition to the minimum inhibitory concentration (MIC) evaluation with an automatic microbiological analyzer. Antibiotic susceptibility was determined based on the Clinical and Laboratory Standards Institute (CLSI) guidelines [26]. In the disc-diffusion method, Muller–Hinton agar was used, and bacterial inoculation was performed according to the CLSI guidelines. The tests using the automatic microbiological analyzer were performed with Gram-positive (Pos MIC Panel type 28, Beckman Coulter, Brea, CA, USA) and Gram-negative (Neg MIC Panel type 44, Beckman Coulter, Brea, CA, USA) antibiotic-susceptibility panels and a combination of the Korean Clinical Practice and CLSI guidelines.

The differences in the related complications and identified bacteria according to the punctal plug types were analyzed using the chi-square test and Fischer’s exact test. Statistical analysis was performed using SPSS^®^ for Windows (version 18.0; IBM Corp., Armonk, NY, USA), and the significance level was set to *p* < 0.05.

## 3. Results

This study was conducted on 58 patients (64 eyes) with a mean age of 57.03 ± 12.9 yr. (range, 23–84 yr.), of which 19 were male and 39 were female. All the patients were Asian. Nine patients were diagnosed with Sjögren syndrome, and six patients had diabetes and hypertension. This study investigated 30 right eyes and 34 left eyes. Among the study participants, six patients had two punctal plugs removed and analyzed: the plugs were removed simultaneously from the lower eyelids of both eyes in three patients, the upper and lower eyelids of the left eye in one patient, the upper eyelids of both eyes in one patient, and the upper eyelid of the left eye and the lower eyelid of the right eye in one patient (Table 1).

Nine eyes of the participants in this study showed an underlying anterior segment eye disease, including Avellino corneal dystrophy, diabetic keratopathy, exposure keratopathy, herpes keratitis, infectious scleritis, keratoconus, nodular episcleritis, Salzmann nodular degeneration, and scleromalacia (Table 1). The most frequent cause of discomfort after insertion of the silicone punctal plug was punctal plug extrusion without granulation (21 eyes, 32.8%), followed by punctal plug extrusion with granuloma (19 eyes, 29.7%), ocular inflammation around the lacrimal punctum (14 eyes, 21.9%), epiphora (9 eyes, 14.1%), and irritation (1 eye, 1.6%). No significant difference was found for the punctal plug types between the SuperEagle Plug™ and Parasol Punctum Plug™ (*p* > 0.05) (Table 2).

The mean interval between insertion and removal of the punctal plug (the retention period) was 7.04 ± 5.7 mo. (range, 0.25–20 mo.), while symptoms of discomfort improved in all patients after removal of the punctal plug. There was no significant correlation between the plug retention period and the incidence of bacterial infection (*p* > 0.05). The punctal plug was removed from the upper eyelid in 14 cases (21.9%) and from the lower eyelid in 50 cases (78.1%), of which 43 cases (67.2%) involved SuperEagle Plug™ insertion and 21 (42.2%) involved Parasol Punctum Plug™ insertion. Among these 64 cases in which the punctal plug was removed, the causal bacteria were identified in the culture test in 27 cases, indicating a bacterial identification rate of 42.2%. In addition, the results of the Gram staining performed alongside the culture test showed that 12 eyes (18.8%) had Gram-positive bacteria and 15 eyes (23.4%) had Gram-negative bacteria (Table 3).

The most frequently identified causative bacteria in the culture test were *Klebsiella aerogenes* in five eyes (18.5%), followed by *Staphylococcus epidermidis* and *Pseudomonas aeruginosa* in 4 eyes (14.8%), and *Citrobacter koseri* in 2 eyes (7.4%). In addition, *Staphylococcus aureus*, *Streptococcus mitis*, *Enterococcus faecalis*, *Bacillus simplex*, *Corynebacterium* species, *Acinetobacter* species, *Enterobacter* species, *Pantoea agglomerans*, and *Stenotrophomonas maltophilia* were identified in one eye each (3.7%). Bacterial isolates showed no significant difference between the SuperEagle Plug™ and Parasol Punctum Plug™ (*p* > 0.05) (Table 4).

In the antibiotic-susceptibility tests performed using the identified bacteria, susceptibility to vancomycin was 100%. In tests performed with quinolone-based antibiotics, the highest susceptibility was 81.0% for levofloxacin, and the lowest susceptibility was 25.0% for moxifloxacin. The overall susceptibility to quinolone-based antibiotics was 73.3%. For cephalosporin-based antibiotics, the highest susceptibility was 93.3% for ceftazidime, followed by cefotaxime (81.8%). In comparisons of the second-, third-, and fourth-generation cephalosporins, the third-generation antibiotics showed the highest susceptibility (88.5%; Table 5).

## 4. Discussion

Silicone punctal plugs can be broadly divided into Freeman and Herrick types [7,27]. The basic structure of the Freeman-type punctal plug consists of a conical head, a cylindrical body, and a cap or collar to fix the plug to the lacrimal punctum [7]. However, the exposed collar can irritate the keratoconjunctival surface or cause plug extrusion and result in complications [9,11,12]. Thus, to reduce the risk of complications, a Herrick-type punctal plug without a collar was developed [27]. Subsequent modifications yielded the silicone punctal plugs that are commonly applied in current clinical practice.

The reported complications of Freeman-type punctal plugs are punctal plug extrusion, epiphora, granuloma formation, irritation of the corneal or conjunctive surface, and canaliculitis [9,11,12]. This study used the SuperEagle Punctum Plug™ and the Parasol Punctum Plug™, which fundamentally resemble the Freeman-type punctual plug, and they caused symptoms such as plug extrusion without granuloma, plug extrusion with granuloma, ocular inflammation around the lacrimal punctum, epiphora, and irritation.

The aforementioned complications of the punctal plug are mostly mild and do not necessitate plug removal [28], and the lacrimal drainage system is generally known to show high tolerance to silicone material [29]. Moreover, silicone material is relatively safe, and complications are often caused by inflammation at the sharp edge or cut side of the silicone or a node rather than being directly induced by the silicone material of the inserted plug [27,30]. The inflammation may also have been caused by the formation of bacterial biofilm on the plug. Bacterial biofilms have been observed even on the punctal plugs of patients who did not show any symptoms of discomfort after plug insertion, and subsequent culture tests confirmed microbial growth in these patients [20]. Since this may lead to opportunistic infection in patients deficient in lactoferrin or lysozymes, which mediate the defense mechanism in the tear [31], biofilm accumulation should be suspected, and the related symptoms should be carefully monitored [32]. The bacteria commonly identified in biofilm-related infections are *Staphylococcus aureus*, *Staphylococcus epidermidis*, and *Pseudomonas aeruginosa* [20]. These strains were likewise identified in this study, but *Klebsiella aerogenes* was the most frequently identified strain. According to one study in South Korea that analyzed the causative bacteria of dacryocystitis, the most commonly detected strains were *Staphylococcus epidermidis*, *Staphylococcus aureus*, and *Klebsiella aerogenes*, the last of which was the most frequently identified Gram-negative bacterial strain in the lacrimal punctum of patients with dacryocystitis [33], while being reported to be the most common strain in chronic dacryocystitis [34]. *Klebsiella aerogenes* is thus considered to be associated with punctal plug or lacrimal drainage. Nonetheless, Jung et al. [33] suggested that in contrast to previous studies, the frequent detection of *Klebsiella aerogenes* should be attributed to the racial or geographical differences of patients with chronic diseases at tertiary hospitals. However, since patients in this study from whom *Klebsiella aerogenes* was identified did not have an underlying disease, the differences are likely to be caused by regional factors. It is also possible that *Klebsiella aerogenes* is the strain associated with punctal plug complications and symptoms of discomfort based on the study by Sugita et al. [20], which included asymptomatic patients in the analysis and reported *Staphylococcus epidermidis* and *Staphylococcus aureus* as the two most frequently detected strains in the given order.

In the previous studies of infectious keratitis in relation to antibiotic susceptibility, Sun et al. [35] reported that the rate of overall resistance to quinolone-based antibiotics was 16.7%, and Kim et al. [36] reported the rate of resistance to ciprofloxacin was 24.3%. In this study, the bacteria cultured from the punctal plug showed an overall susceptibility of 73.3% to quinolone-based antibiotics; thus, the rate of resistance was 26.7%, indicating reduced susceptibility to antibiotics. This is presumed to be due to a greater portion of bacteria with resistance to quinolone-based antibiotics in this study in comparison with previous reports. The antibiotic-susceptibility data in this study showed that the lowest susceptibility (25%) was to moxifloxacin among quinolone-based antibiotics, which is probably due to the small number of tests (*n* = 4 eyes). Most of the cultured bacteria from the punctal plug showed low susceptibility to second-generation cephalosporins (54.5%) and higher susceptibility to the third- and fourth-generation cephalosporins (88.5% and 72.2%, respectively), while the strains resistant to fourth-generation cephalosporins were also resistant to third-generation cephalosporins.

Despite reports describing granulomas with the detection of *Actinomyces* on punctal plugs [37] and granulomas with the detection of *Mycobacterium tuberculosis* after the removal of intracanalicular plugs [38], this study was the first to analyze the different strains of detected bacteria by performing a culture test of the punctal plug removed for a set period of time, which highlights the significance of this study. The limitations, on the other hand, were that the analysis was performed retrospectively using medical records; the sample size was relatively small; the culture test could not be performed for all patients who had undergone silicone punctal plug insertion, and selection bias could not be ruled out since the tests were performed only on patients after the removal of the punctal plug due to discomfort.

This study evaluated the differences in complications and identified bacterial strains for two types of punctal plugs—SuperEagle Punctum Plug™ and Parasol Punctum Plug™—but the differences were not significant (*p* > 0.05). Although the post-insertion retention rate is known to be higher for the Parasol Punctum Plug™ [39], the complications did not show a significant difference in this study, since cases of natural extrusion were not included.

Punctal plug insertion is a simple treatment method that ensures symptomatic improvement for patients with aqueous deficiency dry eye. However, according to a survey conducted by Lee et al. [40], approximately 61% of patients showed adverse reactions after punctal plug insertion, and 51% of these patients received surgical therapy for treating the adverse reactions. Thus, adequate screening of the indications and provision of a thorough description of the positive and adverse effects of punctal plug insertion on the patients is essential before treatment. In patients showing complications, the punctal plug should be removed or replaced, and the cause of the complications should be identified through histological examinations [36] or microbial culture tests to determine an appropriate treatment approach.

## 5. Conclusions

In this study, bacteria were cultured and identified from the punctal plugs removed from approximately 42% of patients who complained of symptoms of discomfort after silicone punctal plug insertion. Thus, microbiological tests may be needed to identify suitable antibiotics for use in punctal plug removal. In addition, the findings of this study are expected to contribute to the selection of antibiotics against causal strains in patients showing complications associated with silicone punctal plugs.

## Figures and Tables

**Table 1 jcm-11-02326-t001:** Demographic features of patients.

	Values
Number of patients (eyes)	58 (64)
Age (years)	57.03 ± 12.9 (23–84)
Gender	
Male	19 (32.8)
Female	39 (67.2)
Laterality	
OD	30 (46.9)
OS	34 (53.1)
Sjögren’s syndrome	9 (15.5)
Diabetes mellitus	6 (10.3)
Hypertension	6 (10.3)
Comorbid anterior segment diseases	9 (15.5)
Avellino corneal dystrophy	1 (1.7)
Diabetic keratopathy	1 (1.7)
Exposure keratopathy	1 (1.7)
Herpes keratitis	1 (1.7)
Infectious scleritis	1 (1.7)
Keratoconus	1 (1.7)
Nodular episcleritis	1 (1.7)
Salzmann nodular degeneration	1 (1.7)
Scleromalacia	1 (1.7)

The values are presented as number (%) or mean ± standard deviation (range).

**Table 2 jcm-11-02326-t002:** Causes of plug removal.

	Number of Isolates (%)	A (%)	B (%)	*p* Value
Protrusion without granuloma	21 (32.8)	12 (27.9)	9 (42.9)	0.232 *
Protrusion with granuloma	19 (29.7)	15 (34.9)	4 (19.0)	0.193 *
Inflammation	14 (21.9)	10 (23.3)	4 (19.0)	0.755 ^†^
Epiphora	9 (14.0)	5 (11.6)	4 (19.0)	0.329 ^†^
Foreign body sensation	1 (1.6)	1 (2.3)	0	1.000 ^†^
Total	64 (100)	43 (100)	21 (100)	

The values are presented as number (%). Group A: SuperEagle Punctum Plug™. Group B: Parasol Punctum Plug™. * Chi-square test; ^†^ Fisher’s exact test.

**Table 3 jcm-11-02326-t003:** Clinical aspects of removed silicone punctal plugs.

	Values
Duration of plug retention (months)	7.04 ± 5.7 (0.25–20)
Plug location	
Upper	14 (21.9)
Lower	50 (78.1)
Kinds of plugs	
SuperEagle Plug™	43 (67.2)
Parasol Punctum Plug™	21 (32.8)
Positive culture results	27 (42.2)
Gram stain	
Gram (+)	12 (18.8)
Gram (−)	15 (23.4)

The values are presented as number (%) or mean ± standard deviation (range).

**Table 4 jcm-11-02326-t004:** Cultured bacterial isolates from removed punctal plugs.

	Number of Isolates (%)	A (%)	B (%)	*p* Value ^†^
Gram (+) cocci				
*Staphylococcus epidermidis*	4 (14.8)	4 (19.0)	0	0.545
*Staphylococcus aureus*	1 (3.7)	1 (4.8)	0	1.000
*Streptococcus mitis*	1 (3.7)	1 (4.8)	0	1.000
*Enterococcus faecalis*	1 (3.7)	1 (4.8)	0	1.000
Gram (+) rod				
*Bacillus simplex*	1 (3.7)	0	1 (16.7)	0.222
*Corynebacterium kroppensteditii*	1 (3.7)	1 (4.8)	0	1.000
*Corynebacterium propinquum*	1 (3.7)	1 (4.8)	0	1.000
*Corynebacterium bovis*	1 (3.7)	1 (4.8)	0	1.000
*Corynebacterium macginleyi*	1 (3.7)	0	1 (16.7)	0.222
Gram (−) rod				
*Klebsiella aerogenes*	5 (18.5)	4 (19.0)	1 (16.7)	1.000
*Pseudomonas aeruginosa*	4 (14.8)	4 (19.0)	0	0.545
*Citrobacter koseri*	2 (7.4)	1 (4.8)	1 (16.7)	0.402
*Acinetobacter species*	1 (3.7)	1 (4.8)	0	1.000
*Enterobacter species*	1 (3.7)	0	1 (16.7)	0.222
*Pantoea agglomerans*	1 (3.7)	0	1 (16.7)	0.222
*Stenotrophomonas maltophilia*	1 (3.7)	1 (4.8)	0	1.000
Total	27 (100)	21 (100)	6 (100)	

The values are presented as number (%). Group A: SuperEagle Punctum Plug™. Group B: Parasol Punctum Plug™. ^†^ Fisher’s exact test.

**Table 5 jcm-11-02326-t005:** Antibiotics susceptibility of cultured bacteria.

Antibiotics	Specimen (*n*)	Susceptible (*n*)	Susceptibility (%)
Fluoroquinolone	45	33	73.3
Ciprofloxacin	20	15	75.0
Levofloxacin	21	17	81.0
Moxifloxacin	4	1	25.0
Vancomycin	6	6	100.0
Cephalosporin			
2nd generation	22	12	54.5
Cefuroxime	13	9	69.2
Cefoxitin	9	3	33.3
3rd generation	26	23	88.5
Ceftazidime	15	14	93.3
Cefotaxime	11	9	81.8
4th generation	18	13	72.2
Cefepime	18	13	72.2

## Data Availability

Data are available upon reasonable request.

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
