# Peer review of "Microbiologic Analysis of Removed Silicone Punctal Plugs in Dry Eye Patients"

_jcm, 2022, doi:10.3390/jcm11092326_

Round 1

Reviewer 1 Report

Punctal plugs are an effective treatment for dry eye disease, one of the most prevalent ocular diseases that affects visual acuity and quality of life. However, a wide spectrum of complications might cause discomfort in patients that require the removal of the plugs. In this manuscript entitled “Microbiologic Analysis of Removed Silicone Punctal Plugs in Dry Eye Patients”, the authors retrospectively examined the microbial profile of the extracted plugs, and compared their antibiotic susceptibility. With dry eye disease being a growing public health concern, this study is a highly clinically relevant piece of work, and may offer some valuable implications for future clinical practice. Overall the manuscript is well written and data appropriately presented, but a few issues listed below need to be addressed.

  1. The interval between insertion and removal of the punctal plugs ranges from 0.25 to 20 months. Is the retention period of the plugs associated with the incidence of bacterial infection?
  2. Among the 64 cases, 7 (10.9%) with Gram-positive bacteria and 12 (18.8%) with Gram-negative bacteria were identified in Table 3. In Table 4, however, 12 cases of Gram-positive and 15 cases of Gram-negative were recorded (consistent with the statement that “causal bacteria were identified in the culture test in 27 cases”-L135). Please clarify the discrepancy.
  3. I’m a bit confused about the statistical analysis for association between bacteria isolates and types of plugs. For example I did a fisher’s test for Staphylococcus epidermidis using a 2 x 2 contingency table (type of plugs vs. bacteria culture outcome) and couldn’t get a p=1.000. I might need some clarification on the hypothesis—were samples that are clean of both Gram-positive and -negative bacteria included for the test?
  4. Dysbiosis of ocular microflora has been demonstrated to contribute to the pathology of dry eye disease. I am curious whether the patients were prescreened for pathogenic bacteria before plug insertion? If pathogens were already present on the ocular surface, would antibiotics treatment alone (without plug insertion) be enough to alleviate the symptoms? Or, would an initial screening of the microflora on patient ocular surface predict the outcome of plug contamination?

Author Response

Point 1: The interval between insertion and removal of the punctal plugs ranges from 0.25 to 20 months. Is the retention period of the plugs associated with the incidence of bacterial infection?

Response 1: There was no significant correlation between the plug retention period and the incidence of bacterial infection, and it was further described in the Line 134-135.

Point 2: Among the 64 cases, 7 (10.9%) with Gram-positive bacteria and 12 (18.8%) with Gram-negative bacteria were identified in Table 3. In Table 4, however, 12 cases of Gram-positive and 15 cases of Gram-negative were recorded (consistent with the statement that “causal bacteria were identified in the culture test in 27 cases”-L135). Please clarify the discrepancy.

Response 2: Thank you for pointing out the discrepancy. We checked the Line 141-142, Table 3 and modified them.

Point 3: I’m a bit confused about the statistical analysis for association between bacteria isolates and types of plugs. For example I did a fisher’s test for Staphylococcus epidermidis using a 2 x 2 contingency table (type of plugs vs. bacteria culture outcome) and couldn’t get a p=1.000. I might need some clarification on the hypothesis—were samples that are clean of both Gram-positive and -negative bacteria included for the test?

Response 3: With samples in which Gram-positive or negative bacteria were detected, Fisher’s test was done again, we found that some p-value measurements were witten incorrectly and corrected them in Table 4.

Point 4: Dysbiosis of ocular microflora has been demonstrated to contribute to the pathology of dry eye disease. I am curious whether the patients were prescreened for pathogenic bacteria before plug insertion? If pathogens were already present on the ocular surface, would antibiotics treatment alone (without plug insertion) be enough to alleviate the symptoms? Or, would an initial screening of the microflora on patient ocular surface predict the outcome of plug contamination?

Response 4: Thank you for your good question. Actually we had not prescreened the patients for pathogenic bacteria before plug inserion. We think that if pathogens were present on the ocular surface, antibiotics therapy alone could possibly resolve the symptoms but as you know, the biofilm on the inserted silicone plugs could have resistance to antibiotics, so the symptoms could persist despite of antibiotics therapy alone and eventually we could need to remove the plugs. We do not think that antibiotics treatment alone could alleviate dry eye symptoms without punctal plug insertion even if pathogens were already present on the ocular surface, because dry eye symptoms could be caused by many other factors. Finally, we agree your opinion: an initial screening of the microflora on patient ocular surface before plug insertion could predict the outcome of plug contamination, so we would include it in the next study.

Reviewer 2 Report

This is a very interesting study regarding possible infection with punctal plug use as they can form biofilms and act as nidus of infection.

  • It is very interesting to see klebsiella as a causative agent and one that would be considered.
  • Very interested to see the susceptibility of moxifloxacin, a commonly used topical drug. This brings to light the issue of antibiotic resistance. 
  • Very well written paper that sheds light on an important topic.

Author Response

Thank you for your meticulous review.

Reviewer 3 Report

Authors have presented a study on microbiologic effects after the removal of silicone punctal plugs from dry eye patients. They found that protrusion without granulation and  protrusion with granuloma are the major causes of for plug removal. They reported that Klebsiella aerogenes is the the most developed microbe and  identified Vancomycin as the best antibacterial agent with highest susceptibility of 100% relative to other agents. Authors concluded that microbiological examination with appropriate selection of antibiotics may be needed since 42% of the patients who reported silicone punctal plugs related discomfort, were identified as infected with  bacterial infection after removing plugs. This is a unique study and the contents are interesting. Overall the manuscript is written well. Authors could incorporate  the following suggestions to improve the visibility of their manuscript

  1. Do authors think that, the presence of other ocular disease (other than dry eye situation) could accelerate  the biofilm deposition ?
  2.  Did authors investigate the studied effects in patients with different ethnicity.
  3. Line 49-57 : Authors could add a short note on the bacterial biofilm detection using non-invasive approaches like optical coherence tomography (OCT) in addition to confocal fluroescence microscopy. ( https://doi.org/10.1117/1.JBO.21.12.127002 https://doi.org/10.1117/12.2190106 ). OCT is a standard clinical tool in ophthalmology that provides in vivo cross sectional and 3D images of retina, anterior segment, and whole eye. This approach has been widely used for bacterial infections in the cornea and other tissues). Authors may add these references and  suggest OCT as a potential approach for the detection of biofilm growth in the studies similar to the one they described in this text. 

Author Response

Point 1: Do authors think that, the presence of other ocular disease (other than dry eye situation) could accelerate the biofilm deposition ?

Response 1: We agree with your idea that the presence of other ocular disease can accelerate the biofilm deposition and described related results of previous studies in the Line 44-48.

Point 2: Did authors investigate the studied effects in patients with different ethnicity.

Response 2: We conducted this study in Asians only and added the content to the Line 104.

Point 3: Line 49-57 : Authors could add a short note on the bacterial biofilm detection using non-invasive approaches like optical coherence tomography (OCT) in addition to confocal fluroescence microscopy. ( https://doi.org/10.1117/1.JBO.21.12.127002 https://doi.org/10.1117/12.2190106 ). OCT is a standard clinical tool in ophthalmology that provides in vivo cross sectional and 3D images of retina, anterior segment, and whole eye. This approach has been widely used for bacterial infections in the cornea and other tissues). Authors may add these references and suggest OCT as a potential approach for the detection of biofilm growth in the studies similar to the one they described in this text.

Response 3: Thank you for the good suggestion. We read the paper you recommended and added related content and reference paper. Please check the Line 51-53 and number 21, 22 in References.